# Nocturia and Sleep in Parkinson’s Disease

**DOI:** 10.3390/jpm13071053

**Published:** 2023-06-27

**Authors:** Ştefania Diaconu, Laura Irincu, Larisa Ungureanu, Diana Țînț, Cristian Falup-Pecurariu

**Affiliations:** 1Department of Neurology, County Clinic Hospital, 500365 Braşov, Romania; 2Faculty of Medicine, Transilvania University, 500036 Braşov, Romania; 3Clinicco, 500059 Braşov, Romania

**Keywords:** nocturia, Parkinson’s disease, urinary tract dysfunctions, sleep dysfunctions, fatigue, non-motor symptoms

## Abstract

Background: Nocturia has a high prevalence in Parkinson’s Disease (PD) and is known to be a bothersome symptom for people with Parkinson’s disease (PwPD). Objective: to characterize nocturia in a sample of PwPD, in relation to sleep, fatigue and other non-motor symptoms (NMS). Methods: we assessed 130 PwPD using a comprehensive battery of scales, which includes the Non-Motor Symptoms Questionnaire (NMSQ), International Parkinson and Movement Disorder Society Non-Motor Rating Scale (MDS-NMS), Parkinson’s Disease Sleep Scale version 2 (PDSS-2), Parkinson’s Disease Questionnaire (PDQ-39), The Overactive Bladder Questionnaire-Short form (OABq-SF), and the Parkinson’s Fatigue Scale (PFS-16). Results: according to the positive answers to the item of the NMSQ related to nocturia, patients were divided into PwPD + nocturia, and PwPD − nocturia. Nocturia was reported by 112 patients (86.15%). Quality of life in PwPD + nocturia was worse than in PwPD − nocturia, according to the PDQ-39 scores (13.32 ± 9.00 vs. 26.29 ± 14.55, *p* < 0.001). Sleep was significantly disturbed in PwPD + nocturia compared to PwPD − nocturia, according to the total scores of various scales, such as PDSS-2, PFS-16. PwPD who complained of nocturia presented higher scores of several NMS. Conclusions: nocturia has a high prevalence in PwPD and it is associated with impaired sleep, fatigue, and reduced quality of life.

## 1. Introduction

Nocturia, defined as voiding during the night, is considered one of the most bothersome complaints among people with Parkinson’s disease (PwPD), with a prevalence varying from 60 to 80% [1,2]. Dysfunctions of the urinary tract are more prevalent in PwPD compared to healthy controls [3]. Urinary dysfunction may be considered a marker of motor and non-motor disability [4]. Urinary symptoms, among other dysfunctions of the autonomic system, may be identified in all stages of the disease, including early Parkinson’s disease (PD) stages [5]. Moreover, urinary dysfunction is a core clinical feature of multiple systemic atrophy (MSA) [6] but is commonly encountered also in progressive supranuclear palsy (PSP) [7]. The existence of dysautonomia and sleep disturbances (especially REM behavior disorder) may resemble PD as PSP-parkinsonism predominant; therefore, neuroimaging examination (magnetic resonance imaging, single-photon emission computerized tomography) might be helpful for the differential diagnosis of atypical parkinsonism, considering the overlap of the clinical characteristics between these entities [8].

Sleep is commonly affected by the presence of nocturia, especially the total sleep time and the sleep efficiency [9]. Among non-motor symptoms, dyssomnia is commonly encountered in PD. The broad spectrum of sleep disorders (including insomnia, excessive daytime sleepiness, restless legs syndrome, REM sleep behavior disorder, sleep disordered breathing, circadian rhythm disturbances) may affect about 60–90% of PwPD; however, therapeutic options for sleep disorders are still limited [10]. Considering the significant negative effects of sleep impairments on daily living and quality of life, identification and assessment of sleep disturbances is mandatory in clinical practice and requires a detailed history taking, along with more comprehensive subjective and objective assessments, including validated scales and questionnaires, or actigraphy and/or polysomnography in complex cases, such as REM sleep behavior disorder [11,12]. Sleep architecture in PD may be impaired even in early stages, concerning both its macrostructure (e.g., fragmented sleep, increased amounts of superficial sleep) and microstructure (related to the perturbation of sleep stages’ characteristics) [13]. Several factors may contribute to the onset and progression of sleep disruption, such as PD-related neurodegeneration (leading to motor and non-motor symptoms, dysautonomia, circadian rhythm impairments, and dysfunction of the ventilatory control), iatrogenic effects of medication, mood disorders (such as depression and anxiety), and sleep syndromes [14].

Motor symptoms during the night (akinesia, dyskinesia, or tremor) and non-motor symptoms (such as pain and nocturia) represent important factors related to sleep fragmentation [11]. Screening for nocturia therefore represents an important component of the personalized assessment of patients with insomnia [15]. Reduced bladder capacity, detrusor overactivity, and the loss of inhibition of the voiding reflex are the main pathophysiological mechanisms that explain the common occurrence of nocturia in PwPD [16]. The altered function of the frontal cortex–basal ganglia circuit might contribute to the abnormal overactivity of the detrusor muscle and lead to urinary urgency, frequent micturition, and incontinence [17]. Single-photon emission computerized tomography (SPECT) studies have demonstrated that the depletion of dopaminergic nigrostriatal neurons correlates with the severity of bladder dysfunction [18,19].

The aims of this study were to determine the prevalence of nocturia in Parkinson’s disease, and to explore the characteristics of nocturia in relation to sleep, fatigue, and other non-motor symptoms (NMS).

## 2. Materials and Methods

### 2.1. Patients and Study Design

This cross-sectional study enrolled 130 people with Parkinson’s disease, and was conducted within the Department of Neurology of the County Clinic Hospital, Transilvania University of Braşov. The inclusion criteria were people with a diagnosis of PD (based on the criteria developed by the Movement Disorders Society (MDS) [20]; absence of severe cognitive impairments (based on Mini Mental State Examination Test); and signed informed consent. The exclusion criteria were secondary and atypical forms of parkinsonism; schizophrenia and bipolar disorders; severe bladder or prostate disease, or any neurological causes of bladder dysfunction; and history of pelvic or prostatic surgery.

### 2.2. Clinical Assessments

A comprehensive case report form was applied in order to assess urinary function in relation to sleep and other non-motor symptoms. It contained information regarding demographic data (age, gender, age at PD onset, subtype of PD, levodopa equivalent daily dose (LEDD), duration of the disease, Hoehn and Yahr (H&Y) stages, other associated medical conditions, and treatment and various validated scales which will be described below. The assessment of the motor function was performed in “ON” states by a specialist in neurology. All participants signed the informed consent forms. This study was approved by the Ethics Committee of University Transilvania of Braşov (1.11/01/2019) and it was conducted in accordance with the principles of the Declaration of Helsinki.

### 2.3. Questionnaires and Rating Scales

Motor characteristics and the severity of PD were evaluated according to the MDS Unified Parkinson’s Disease Rating Scale part III (MDS-UPDRS III) [21], Scales for Outcomes in Parkinson’s disease (SCOPA)—the Motor scale [22], and H&Y staging. The NMS of the patients were evaluated with the Non-Motor Symptoms Questionnaire (NMSQ) [23], which consists of 30 items regarding the presence of several NMS, and International Parkinson and Movement Disorder Society-Non-Motor Rating Scale (MDS-NMSS) [24], in which the frequency and the severity of the NMS is evaluated, together with some questions related to NMS fluctuation.

Based on the answer ‘yes’ to the item number 9 of the NMSQ, ‘Getting up regularly at night to pass urine’, patients were divided into two groups—with or without nocturia (PwPD + nocturia, and PwPD − nocturia, respectively). Later on, patients were evaluated using The Overactive Bladder Questionnaire-Short Form (OABq-SF), which is a reliable, validated tool designed for the brief self-assessment of urinary symptoms and their impact on quality of life (QoL) over the past 4 weeks. It consists of two scales: one Symptom Bother scale, with 6 items, and one Health-Related Quality of Life scale, comprising 13 items [25]. For all the items, the patient chooses one answer from six degrees of the severity/frequency of the symptoms’ occurrence. The OABq-SF is classified by the MDS Task Force as a “recommended with caveats” tool for evaluating urinary function in PwPD [26].

Sleep was assessed using various scales. The Parkinson Disease Sleep Scale (PDSS)-2 [27] is a self-assessment rating tool of the frequency of various sleep disturbances, and it was designed specifically for the PD population. Scores higher than 15 were proposed to help identifying “bad sleepers” [27]. The Parkinson’s Disease Questionnaire (PDQ-39) [28] is a scale used commonly for assessing the QoL of the PwPD. The Insomnia Severity Index (ISI) [29] is a validated tool used for the evaluation of the night time and daytime characteristics of insomnia. Another instrument for evaluating insomnia is the Athens Insomnia Scale (AIS), which consists of eight items regarding the components of insomnia (first five questions) and their consequences (last three items) [30]. A total AIS score >6 suggests insomnia [31]. Excessive daytime sleepiness (EDS) was evaluated with the Epworth Sleepiness Scale (ESS) [32], which is a widely used tool for self-estimation of sleepiness, based on the probability of falling asleep in some certain described situations.

We evaluated fatigue using the Parkinson Fatigue Scale (PFS-16) [33]. This scale was developed specifically for the PD population, and contains 16 items in which fatigue and its consequences are self-assessed by the patients. The responses for each item can range from “1: strongly disagree” to “5: strongly agree”. Fatigue was identified in cases in which the total mean score was >3.3, according to previous studies [33].

For cognitive assessment, we used the Mini-Mental State Examination (MMSE) and Montreal Cognitive Assessment (MoCA), the latter being indicated in assessing cognitive function in patients with neurodegenerative disorders [34,35]. Anxiety and depression were self-assessed by PwPD using the Hospital Anxiety and Depression Scale (HADS), which consists of two scales (for depression and for anxiety, respectively). Higher values are suggestive of worse symptoms [36].

### 2.4. Data Analysis

Statistical analyses were performed with IBM SPSS Statistics for Windows, Version 26.0. Armonk, NY, USA: IBM Corp. Data were expressed as means ± standard deviation (SD). Probability values of *p* < 0.05 were considered significant. The sample distribution was determined using the Shapiro–Wilk test. Non-parametric statistics were used because the main variables in the study did not meet assumptions for use of parametric tests. To determine the relationship between sleep scores and other NMS, Spearman rank correlation coefficients were calculated. Fisher’s exact test and the Mann–Whitney *U* Test were used to compare the characteristics of PwPD between groups (with or without nocturia). A logistic regression model was performed in order to determine the predictors of nocturia.

## 3. Results

In our study, 130 consecutive PwPD were enrolled, 68 (51.9%) of them being females. Based on item 9 of the NMSQ related to nocturia, 112 patients (86.15%) reported regular nocturia. Nocturia was reported more commonly by women (*p* = 0.041); except for this finding, no other significant differences were observed regarding age, years of evolution of the disease, or LEDD between PwPD + nocturia and PwPD − nocturia (Table 1).

Significant differences were noted in between groups related to motor symptoms (UPDRS III total score: 33.96 ± 13.85 in PwPD + nocturia vs. 24.55 ± 8.8 in PwPD − nocturia, *p* = 0.004; SCOPA Motor: 23.25 ± 9.07 in PwPD + nocturia vs. 17.88 ± 6.52 in PwPD − nocturia, *p* = 0.023). No significant difference was noted regarding the H&Y stage between groups (Table 2). Higher scores of OABq-SF total, OABq-SF Symptom bother and OABq-SF Health-Related QoL scales were observed in the PwPD + nocturia group compared to the PwPD − nocturia group.

QoL in PwPD + nocturia was worse than in PwPD − nocturia, according to the PDQ-39 total scores (13.32 ± 9.00 vs. 26.29 ± 14.55, *p* < 0.001). Sleep was significantly more impaired in PwPD + nocturia compared to PwPD − nocturia, according to the total scores of various scales: PDSS-2, ISI, AIS, ESS, PFS-16 (Table 2). Patients complaining of nocturia also presented more excessive daytime sleepiness, as demonstrated by higher ESS scores (*p* = 0.006).

Maintenance insomnia (as suggested by item 2 of ISI: “Difficulty staying asleep” and item 2 of AIS “Awakening during the night”) correlated with the presence of nocturia (rho 0.366, and 0.428, respectively). The positive answer for nocturia (questions 4 and 5 of the OABq-SF, related to night-time urination and waking up at night due to the need to urinate) significantly correlated with fatigue, insomnia and worse quality of life (according to the total scores of PFS-16, ISI, and PDQ-39, *p* < 0.05)

Various sleep problems could be identified in PwPD + nocturia, when evaluating the individual components of PDSS-2. Compared to patients without nocturia, PwPD + nocturia presented worse scores related to difficulty falling asleep and maintaining sleep, distressing dreams, distressing hallucinations, discomfort related to akinesia and immobility, pain in arms and legs, muscle cramps, painful postures of arms and legs upon waking up, tiredness and sleepiness in the morning, snoring, and breathing difficulties (Figure 1).

We further classified PwPD as “bad sleepers” if the PDSS-2 score was over 15, and “fatigued” if the mean score of PFS-16 was over 3.3. We analyzed if sleep problems or fatigue are correlated with more severe urinary problems, as evaluated with OABq-SF. Even if not statistically significant, patients identified as bad sleepers or fatigued were found to have higher scores of OABq SF-total score, OABq SF-symptom subscale and OABq SF-QoL subscale.

No differences between groups were noted in terms of cognition, as evaluated with the MMSE and MoCA tests. We observed higher scores of HADS total score and sub-scores (HADS depression and HADS anxiety) in PwPD + nocturia. Significantly higher total scores of MDS-NMSS were noted in the PwPD + nocturia group (*p* < 0.001)—Table 2.

Furthermore, nocturia was correlated with several other motor and non-motor features, as presented in Table 3. Significant correlations were observed between nocturia and total scores of MDS-UPDRS part III, SCOPA Motor, PDQ-39, ISI, AIS, PDSS, ESS, NMSQ, MDS-NMS, HADS (and subscales HADS depression and HADS anxiety), OABq-SF total, OABq-SF Symptom bother, and OABq-SF Health-Related QoL.

In order to characterize nocturia in relation to other non-motor symptoms, we further compared the differences regarding the domains of MDS-NMSS in the two groups. In comparison with patients without nocturia, patients who reported nocturia presented higher scores in the following domains: depression, anxiety, apathy, cognition, urinary, gastrointestinal, sleep and wakefulness, pain, and miscellaneous (Table 4).

Within the logistic regression model (Table 5), the MDS-UPDRS III total score, depression, anxiety, urinary, gastrointestinal, sleep and wakefulness, pain, and miscellaneous can predict whether the patient suffers from nocturia (as defined by the yes/no answer to the item number 9 of the NMSQ).

## 4. Discussion

The prevalence of nocturia in our study group was high (86.15%), in line with the results of other studies [1]. On the contrary, there were also studies reporting a lower prevalence. One example is the PRIAMO multicentric study, in which nocturia was reported in 34.6% of the PwPD, based on the NMSQ assessment [37]. These discrepancies could be related to the variability of the methods used to assess nocturia, the inhomogeneity of the studied sample (demographic features, comorbidities, etc.) and differences in stages of enrolled PwPD in different studies. Furthermore, nocturia is more prevalent in PwPD compared to healthy controls. One questionnaire-based study on 61 PwPD and 74 healthy controls observed that almost 64% of the PwPD reported urinary symptoms, compared to 32.8% of the subjects in the control group [38]. Using the Overactive Bladder Questionnaire (OAB-q), Iacovelli et al. observed significantly higher scores of OAB-q in PwPD compared to healthy individuals, and correlations between the OAB-q scores, age, disease duration and motor severity [39].

In our study group, nocturia was reported more commonly by females. We did not observe any associations between nocturia and other demographic characteristics, such as age, PD duration or LEDD. Similar to our results, one study reported a prevalence of nocturia of 56% of men and 63% of women in PwPD [40]. On the contrary, another study involving 110 PwPD observed similar results between males and females regarding age, disease duration and severity, and impact on QoL [41].

In our study group, patients with nocturia presented a worse motor status, as evaluated with MDS-UPDRS III and SCOPA-motor scale. The occurrence of nocturia was associated with severity of PD in previous studies, when considering motor assessments [42,43,44]. The study conducted by Xu et al., in which 100 PwPD were included, reported that nocturia (evaluated with the Overactive Bladder Symptom Score—OBSS) is associated with older age and higher H&Y stages [45]. Using the same assessment scale, Mito et al. noticed a significant correlation between the severity of nocturia and a more advanced motor stage, according to the UPDRS—III evaluation [46]. In vitro studies have demonstrated that dopamine-mediated impulses from substantia nigra pars compacta have an important role on the inhibition of the voiding reflex, via D1 receptors [47]; therefore, the degeneration of the dopaminergic neurons found in PD in substantia nigra and the impairment of the cortical–basal ganglia circuitry may cause detrusor hyperreflexia and the occurrence of nocturia [17,48]. As the neurodegeneration process evolves, urinary symptoms may worsen over time due to impaired control of the voiding reflex.

Nocturia seems to have an important impact on quality of life and quality of sleep. As expected, patients who were identified as having nocturia (based on NMSQ evaluation) had higher scores on OABq-SF total, OABq-SF Symptom bother and OABq-SF Health-Related QoL, suggesting more impairments of the urinary tract in these patients and a worse QoL. We found correlations that were statistically significant between nocturia and high scores of PDQ-39 (suggesting a reduced quality of life) and high PDSS-2 total scores (representative for sleep disturbances). Several subdomains of the PDSS-2 were impaired in patients with nocturia, such as difficulty initiating and maintaining sleep, immobility, and tiredness and sleepiness in the morning. Hallucinations may have a high prevalence in PD and might be related to disrupted circuitry, especially at the level of the thalamus [49]. We consider that the association we have found between nocturia and distressing dreams or hallucinations is an interesting aspect which was unexplored in previous studies. Patients with nocturia present more excessive daytime sleepiness and more insomnia (especially maintenance insomnia). Moreover, “bad sleepers” and fatigued patients tend to have more severe urinary disturbances compared with patients that do not present sleep or fatigue.

In the general population, two or more voids per night were considered to be bothersome, according to a survey involving 6000 participants [50]. A meta-analysis stated that seven factors are directly related to nocturia: disease duration, age, sleep impairments, constipation, H&Y scores >2, and higher scores on the UPDRS-III and MMSE evaluations [51]. The strong association between nocturia and sleep impairments may be explained by the delayed sleep re-onset after awakening during night due to the need to void [52]. Difficulties in falling asleep lead to fatigue in the morning, and are considered bothersome by the patients [53]. One questionnaire-based study reported that total scores of the Pittsburgh Sleep Quality Index were higher in PwPD + nocturia, suggesting a worse quality of sleep in these patients [44]. One polysomnographic study has demonstrated that nocturia is associated with disrupted sleep, and reduced total sleep time and sleep efficiency; therefore, PwPD with high bother due to nocturia tend to have a worse sleep quality [9].

It was proven that nocturia has significant correlations with other NMS. One study in which 314 PwPD were enrolled reported a significant association between nocturia and anxiety, which was more prominent in male patients. The authors suggest that multiple instances of awakening during the night due to the need to void induce a state of distress for these patients, and contribute to a vicious circle in which nocturia, anxiety, interrupted and inadequate sleep, and fear of falling are involved [54]. According to Benli et al., lower urinary dysfunction may be a risk factor for the development of anxiety and depression [55]. According to one study, the number of voids/night was positively correlated to depression, anxiety, sleep quality, and the severity of PD [44]. Similar to the results of previous studies, we found significant correlations between nocturia and depression and anxiety.

When evaluating nocturia in relation to cognition, we did not observe any objective association between the total scores of MMSE or MoCA and nocturia. However, an important correlation was found between nocturia and the total score of the ‘F. Cognition domain’ of MDS-NMSS. We believe that this discrepancy could be explained by the subjective insight regarding cognitive problems that can be different from the objective assessment tests. There were several studies that have reported an association between urinary dysfunction and cognitive impairments [45,56], independent of PD duration or medication [56]. One explication for this association can be drawn based on the neurodegeneration of the prefrontal cortex, which has important roles in controlling both micturition [57] and executive functions [45,58]. Other non-motor domains that presented correlations with nocturia in our study were apathy, pain, and gastrointestinal domains. According to the results of the meta-analysis conducted by Zhuang et al., constipation represents a risk factor for dysfunction of the lower urinary tract [51].

To the best of our knowledge, there are no previous studies exploring the relationship between nocturia and fatigue in PwPD. In the general population, nocturia, as an important cause of sleep disruption and sleep deprivation, is also associated with fatigue during the day; moreover, the higher the number of voids/night, the more fatigue and impaired sleep can be observed in general population [59].

One study evaluated autonomic dysfunction in PD in relation to fatigue in 94 PwPD using, among other scales, the SCOPA-Autonomic Questionnaire (SCOPA-AUT) and the Fatigue Severity Scale (FSS). The dysautonomic features related to the urinary, gastrointestinal and thermoregulatory domains were strongly correlated with fatigue. However, the authors do not mention whether this correlation was observed solely for the urinary domain or specifically for its individual compounds (including nocturia) [60]. Similar results were obtained by Zhao et al. in a study including 602 PwPD. Correlations were found between fatigue (as assessed with FSS) and other autonomic domains, including the urinary domain (as evaluated using the Non-Motor Symptoms Scale and SCOPA-AUT) [61]. In a subgroup of PwPD in the PRIAMO study, various non-motor symptoms (NMS) were associated with urinary dysfunction, such as gastro-intestinal symptoms, sleep, pain, and fatigue [4]. Nocturia was not assessed separately in any of these studies. The co-existence of fatigue with other autonomic dysfunctions may be explained by the Lewy bodies’ accumulation in the lower raphe nuclei and the reticular formation [62].

Our study has several limitations, two of which are the small number of participants and the lack of a control group, which would have been useful for proper comparisons between groups. This is a questionnaire-based study, and we did not assess the patients with objective tools such as polysomnography or actigraphy. The participants’ urinary function was not assessed by an urologist; we did not use charts to evaluate frequency and flow void/24 h or other objective diagnostic tools for the urinary symptoms.

## 5. Conclusions

Nocturia is a common symptom among people with Parkinson’s disease; it has important consequences, considering its association with sleep disruption, reduced QoL, depression, anxiety, daytime sleepiness, and fatigue. Further research, which should also include longitudinal assessments and objective evaluation of nocturia in relation to sleep parameters (e.g., urodynamic studies, polysomnography), is necessary for establishing to what extent these features are interrelated, and to develop novel personalized therapeutic strategies, which should better adapt to the patients’ needs.

## Figures and Tables

**Figure 1 jpm-13-01053-f001:**
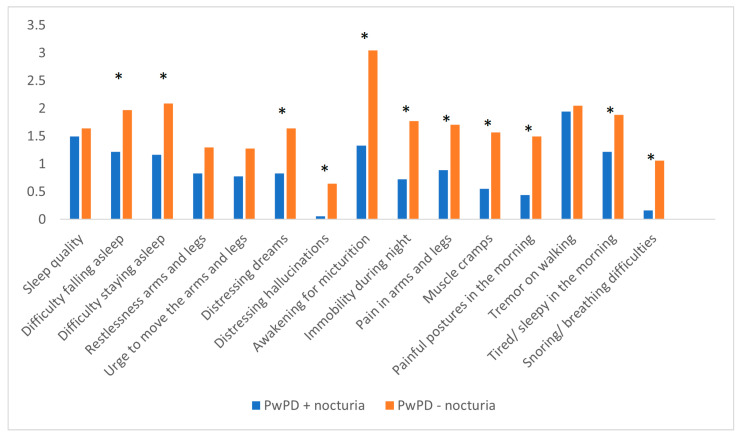
PDSS-2 scores of individual items in PwPD + nocturia vs. PwPD − nocturia. *Significant differences were marked with asterisk (*).* PDSS-2: Parkinson Disease Sleep Scale-2; PwPD: people with Parkinson’s disease.

**Table 1 jpm-13-01053-t001:** Demographic data of PwPD and nocturia vs. PwPD without nocturia (Fisher’s exact test).

	PwPD + Nocturia (n = 112)	PwPD − Nocturia (n = 18)	*p*-Value
Sex (M/F)	54/58	8/10	**0.041**
LEDD (mg, mean ± SD)	508.35 ± 399.95	451.38 ± 277.75	0.523
Age (years ± SD)	74.77 ± 8.97	72.61 ± 9.17	0.485
Age at PD onset (years ± SD)	69.32 ± 10.21	69.38 ± 9.86	0.178
Duration of PD (years ± SD)	5.35 ± 4.40	3.47 ± 2.36	0.877

LEDD: Levodopa equivalent daily dose; PD: Parkinson’s disease; PwPD: people with Parkinson’s disease; SD: standard deviation. Bold values denote statistical significance (*p* < 0.05).

**Table 2 jpm-13-01053-t002:** Main characteristics of motor and non-motor symptoms in the two groups (Mann–Whitney *U* Test).

	PwPD + Nocturia	PwPD − Nocturia	*p*-Value
**Motor symptoms**			
H&Y (mean ± SD)	2.44 ± 0.76	2.11 ± 0.58	0.103
MDS-UPDRS III (mean ± SD)	33.96 ± 13.85	24.55 ± 8.8	**0.004**
SCOPA Motor (mean ± SD)	23.25 ± 9.07	17.88 ± 6.52	**0.023**
**Non-motor symptoms**			
PDQ-39	26.29 ± 14.55	13.32 ± 9.00	**<0.001**
AIS	8.99 ± 4.94	4.28 ± 3.77	**<0.001**
ISI	11.38 ± 5.91	5.50 ± 4.42	**<0.001**
PDSS-2	25.17 ± 10.56	13.67 ± 8.18	**<0.001**
ESS	9.52 ± 5.51	5.67 ± 4.14	**0.006**
MMSE	27.21 ± 3.32	27.83 ± 3.80	0.136
MoCA	23.30 ± 5.35	24.28 ± 6.69	0.156
HADS Depression	9.33 ± 4.31	6.44 ± 3.91	**<0.001**
HADS Anxiety	5.73 ± 3.84	3.27 ± 2.80	**0.009**
HADS Total	15.06 ± 7.71	9.72 ± 6.37	**0.007**
PFS-16	3.20 ± 1.09	2.39 ± 1.06	**0.006**
OABq-SF total	40.82 ± 18.22	22.06 ± 2.777	**<0.001**
OABq-SF Health-related QoL	25.37 ± 11.27	14.28 ± 1.40	**<0.001**
OABq-SF Symptoms bother	15.46 ± 7.39	7.78 ± 1.92	**<0.001**
NMSQ TOTAL	10.44 ± 5.00	5.17 ± 3.06	**0.014**
MDS-NMSS TOTAL	62.45 ± 46.67	25.16 ± 21.21	**<0.001**

AIS: Athens Insomnia Scale; ESS: Epworth Sleepiness Scale; H&Y: Hoehn and Yahr; HADS: Hospital Anxiety and Depression Scale; ISI: Insomnia Severity Index; MDS-NMSS: Movement Disorder Society-Non-Motor Rating Scale; MDS-UPDRS: Movement Disorders Society Unified Parkinson’s Disease Rating Scale; MMSE: Mini-Mental State Examination; MoCA: Montreal Cognitive Assessment; NMSQ: Non-Motor Symptoms Questionnaire; OABq-SF: Overactive Bladder Questionnaire-Short form; PD: Parkinson’s disease; PDQ-39: Parkinson’s Disease Questionnaire; PDSS-2: Parkinson Disease Sleep Scale-2; PFS-16: Parkinson Fatigue Scale; PwPD: people with Parkinson’s disease; RLS: restless legs syndrome; SCOPA: Scales for Outcomes in Parkinson’s disease. Bold values denote statistical significance (*p* < 0.05).

**Table 3 jpm-13-01053-t003:** Correlations between nocturia (answer ‘yes’ to the item 9 of the NMSQ) and several other motor and non-motor features (Spearman rank correlation).

NMSQ Item 9	Correlation Coefficient	*p*-Value
Sex	−0.069	0.436
Age (years ± SD)	0.097	0.270
Age at PD onset (years ± SD)	−0.033	0.707
Duration of PD (years ± SD)	0.142	0.108
LEDD	0.004	0.965
H&Y (ON)	0.144	0.103
MDS-UPDRS III total score	0.251	**0.004**
SCOPA-MOTOR total score	0.202	**0.023**
PDQ-39 total score	0.327	**<0.001**
ISI total score	0.338	**<0.001**
AIS total score	0.322	**<0.001**
PDSS-2 total score	0.357	**<0.001**
ESS total score	0.241	**0.006**
NMSQ total score	0.349	**0.014**
MDS-NMS total score	0.340	**<0.001**
MMSE total score	−0.131	0.137
MoCA total score	−0.125	0.157
HADS total score	0.248	**0.004**
HADS Anxiety total score	0.232	**0.008**
HADS Depression total score	0.239	**0.006**
PFS 16	0.243	**0.005**
OABq-SF Symptom bother total score	0.451	**<0.001**
OABq-SF Health-related QoL total score	0.497	**<0.001**
OABq-SF TOTAL	0.498	**<0.001**

AIS: Athens Insomnia Scale; ESS: Epworth Sleepiness Scale; H&Y: Hoehn and Yahr; HADS: Hospital Anxiety and Depression Scale; ISI: Insomnia Severity Index; MDS-NMSS: Movement Disorder Society-Non-Motor Rating Scale; LEDD: Levodopa equivalent daily dose; MDS-UPDRS: Movement Disorders Society Unified Parkinson’s Disease Rating Scale; MMSE: Mini-Mental State Examination; MoCA: Montreal Cognitive Assessment; NMSQ: Non-Motor Symptoms Questionnaire; OABq-SF: Overactive Bladder Questionnaire-Short form; PD: Parkinson’s disease; PDQ-39: Parkinson’s Disease Questionnaire; PDSS-2: Parkinson Disease Sleep Scale-2; PFS-16: Parkinson Fatigue Scale; PwPD: people with Parkinson’s disease; RLS: restless legs syndrome; SCOPA: Scales for Outcomes in Parkinson’s disease; SD: standard deviation. Bold values denote statistical significance (*p* < 0.05).

**Table 4 jpm-13-01053-t004:** Comparison between PwPD and nocturia vs. PwPD without nocturia in relation to MDS-NMSS domains.

	PwPD + Nocturia	PwPD − Nocturia	*p*
MDS-NMSS A. Depression	3.69 ± 4.88	1.00 ± 1.32	**0.005**
MDS-NMSS B. Anxiety	3.37 ± 4.56	0.72 ± 1.56	**0.002**
MDS-NMSS C. Apathy	2.21 ± 3.39	0.72 ± 1.48	**0.019**
MDS-NMSS D. Psychosis	0.31 ± 0.96	0.11 ± 0.32	0.653
MDS-NMSS E. Impulse control disorders	0.02 ±0.21	0.00 ± 0.00	0.569
MDS-NMSS F. Cognition	8.39 ± 11.97	3.83 ± 6.74	**0.028**
MDS-NMSS G. Orthostatic hypotension	3.35 ± 5.41	1.72 ± 3.62	0.110
MDS-NMSS H. Urinary	8.02 ± 8.69	1.22 ± 1.06	**<0.001**
MDS-NMSS I. Sexual	0.15 ± 0.95	0.00 ± 0.00	0.363
MDS-NMSS J. Gastrointestinal	2.50 ± 3.35	0.72 ± 1.31	**0.023**
MDS-NMSS K. Sleep and wakefulness	11.00 ± 8.28	4.94 ± 5.96	**<0.001**
MDS-NMSS L. Pain	9.17 ± 8.29	3.66 ± 4.48	**0.002**
MDS-NMSS M. Miscellaneous	10.21 ± 7.33	6.50 ± 6.84	**0.023**

MDS-NMSS: Movement Disorder Society-Non-Motor Rating Scale; PwPD: people with Parkinson’s disease. Bold values denote statistical significance (*p* < 0.05).

**Table 5 jpm-13-01053-t005:** Logistic regression analysis for determining the impact of different non-motor symptoms on nocturia (item number 9 of the NMSQ).

			95% C.I. for OR	
	*p*	OR	Lower	Upper
MDS-UPDRS III TOTAL	**0.009**	1.073	1.018	1.132
MDS-NMS A. Depression	**0.033**	1.482	1.031	2.129
MDS-NMS B. Anxiety	**0.028**	1.484	1.043	2.111
MDS-NMS H. Urinary	**0.001**	4.338	1.826	10.307
MDS-NMS J. Gastrointestinal	**0.043**	1.417	1.010	1.986
MDS-NMS K. Sleep and wakefulness	**0.005**	1.183	1.052	1.330
MDS-NMS L. Miscellaneous	**0.011**	1.164	1.036	1.308

NMSQ: Non-Motor Symptoms Questionnaire; MDS-NMSS: Movement Disorder Society-Non-Motor Rating Scale; MDS-UPDRS: Movement Disorders Society Unified Parkinson’s Disease Rating Scale; OR: odds ratio; CI: confidence interval; Bold values denote statistical significance (*p* < 0.05).

## Data Availability

The data presented in this study are available on reasonable request from the corresponding author.

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
