# Peer review of "Nocturia and Sleep in Parkinson’s Disease"

_jpm, 2023, doi:10.3390/jpm13071053_

Round 1

Reviewer 1 Report

The authors of this paper explored the prevalence factor associated with nocturia in PD. They assessed 130 patients in a cross-sectional manner. The methodology employed and the results are sound. I suggest that the association between nocturia and other NMS should include at least gender and the motor severity score as confounding factors, by means of a multivariate technique.

It is ok. I only detected a few minor typos.

Author Response

We thank the reviewer for the comments. The answers can be found in the attached document.

Reviewer 2 Report

Authors present the significance of nocturia in Parkinson's Disease (PD). Sleep abnormalities in parkinsonism are a significant feature, however insufficiently described. I have several suggestions concerning the work:

1. In the introduction authors should also refer to sleep abnormalities, dysautonomia in other clinical entities possibly resembling PD as PSP-P.

Ref. 

The Strengths and Obstacles in the Differential Diagnosis of Progressive Supranuclear Palsy-Parkinsonism Predominant (PSP-P) and Multiple System Atrophy (MSA) Using Magnetic Resonance Imaging (MRI) and Perfusion Single Photon Emission Computed Tomography (SPECT). Diagnostics (Basel). 2022 Feb 2;12(2):385. doi: 10.3390/diagnostics12020385. PMID: 35204476; PMCID: PMC8871165.

2. Was the subtype of PD patients verified?

3. Exclusion criteria - were any psychiatrical entities indicated as exclusion criteria?

4. Authors should provide an additional paragraph on dyssomnia in parkinsonisms

5. The conclusion section is vague, does not provide a sufficient overview of future perspectives.

Author Response

(The authors gave the same response as above.)

Round 2

Reviewer 1 Report

Thanks for adding the requested analysis. Please consider the following proposal: change the multiple regression for a logistic regression analysis to predict nocturia (yes vs. no), including the following variables: depression, anxiety, apathy, cognition, urinary, gastrointestinal, sleep and wakefulness, pain, and miscellaneous, LEDD, MDS-UPDRS III score. You can add a table with ORs (95% CI). This analysis will also allow you to see which of these variables are independently connected with nocturia. 

Author Response

We thank you for the suggestion. Please see the answer in the attachment.

Reviewer 2 Report

I have no further comments.

Author Response

We thank you very much.